# Canopy Assessment of Cycling Routes: Comparison of Videos from a Bicycle-Mounted Camera and GPS and Satellite Imagery

**Albert Bourassa [1,2]**, **Philippe Apparicio [1,\*]**, **Jérémy Gelb [1]** and **Geneviève Boisjoly [3]**

1   Institut National de la Recherche Scientifique, Centre Urbanisation Culture Société, 385 Sherbrooke E, Montréal, QC H2X 1E3, Canada
2   Département d'études Urbaines et Touristiques, Université du Québec à Montréal, 315 rue Sainte-Catherine E, Montréal, QC H2X 3X2, Canada
3   Département des Génies Civil, Géologique et des Mines, Polytechnique Montréal, 2500 Chem. de Polytechnique, Montréal, QC H3T 1J4, Canada
\*   Correspondence: philippe.apparicio@inrs.ca

**Abstract:** Many studies have proven that urban greenness is an important factor when cyclists choose a route. Thus, detecting trees along a cycling route is a major key to assessing the quality of cycling routes and providing further arguments to improve ridership and the better design of cycling routes. The rise in the use of video recordings in data collection provides access to a new point of view of a city, with data recorded at eye level. This method may be superior to the commonly used normalized difference vegetation index (NDVI) from satellite imagery because satellite images are costly to obtain and cloud cover sometimes obscures the view. This study has two objectives: (1) to assess the number of trees along a cycling route using software object detection on videos, particularly the Detectron2 library, and (2) to compare the detected canopy on the videos to other canopy data to determine if they are comparable. Using bicycles installed with cameras and GPS, four participants cycled on 141 predefined routes in Montréal over 87 h for a total of 1199 km. More than 300,000 images were extracted and analyzed using Detectron2. The results show that the detection of trees using the software is accurate. Moreover, the comparison reveals a strong correlation (>0.75) between the two datasets. This means that the canopy data could be replaced by video-detected trees, which is particularly relevant in cities where open GIS data on street vegetation are not available.

**Keywords:** bicycle; trees; canopy; NDVI; video; automatic detection; Detectron2; Montréal

## 1. Introduction

Connecting with nature provides many health benefits [1–3]. A great way of improving mental and physical health is by using a bicycle to travel and commute [4]. However, many factors such as the built environment affect the decision to use a bicycle as a mode of transportation [5,6]. According to Winters et al. [7], when choosing a route to cycle, beautiful scenery is the second most important motivator in a person's decision after routes that minimize exposure to noise and air pollution and before cycling paths that are separated from road traffic. Determining the greening of cities, particularly the abundance of street trees, can therefore be a tool for understanding travel behavior or assessing the quality of bicycle routes in a city. Indeed, the tree canopy is an element of nature that contributes to positive emotions [8] and nicer scenery [9]. Routes with trees are also preferred by pedestrians and cyclists [10].

A popular method of determining greenness in cities is to calculate the normalized difference vegetation index (NDVI) from satellite imagery [11,12]. Although interesting, this approach has two significant drawbacks: (1) high-resolution imagery can be costly to obtain, and (2) sometimes, parts of the images are not usable due to cloud cover.

An alternative is to analyze street view images—obtained from Google Street View (GSV) or BMap—which provide a visualization of vegetation as seen by an individual on

the street [13–16]. For example, based on GSV, Li et al. [14] proposed a green view index (GVI) that varies from 0 to 100 for the percentage of street vegetation in city scenery. In this respect, a recent Canadian study based on a survey of 282 adults found a significant positive association between the GVI and participation in recreational activities during the summer, whereas no significant association was found with the NDVI [15]. However, the use of GSV has two important limitations: (1) the street data collection of a whole city can be carried out over several seasons, including winter (Figure 1), or even years, and (2) GSV images have not been captured everywhere in the world, especially in cities in the Global South.

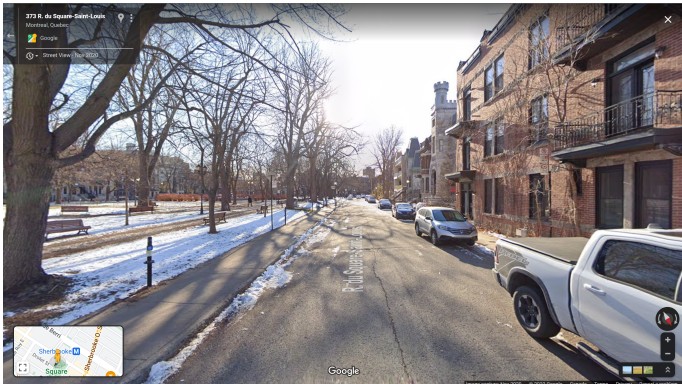

**Figure 1.** Google Street View image for a road in Montreal in 2020.

Over the last two decades, video cameras have become a popular tool for recording and analyzing data in the field [17,18] including in transportation studies [19–23]. New algorithms using artificial intelligence can detect the components in an image to identify the objects including features in the built environment. The most popular libraries for such use are Detectron2, EfficientDet, YOLO, and Faster R-CNN, with Detectron2 being the most accurate [24]. Furthermore, Detectron2 is already being used in transportation studies to count the number of cars on a highway [25]. To the best of our knowledge, there are, however, no studies that currently use such algorithms to determine if videos taken in a city can accurately show the amount of greenness on different routes and road types. These videos taken at eye level with a camera mounted on a bicycle handlebar could also provide new real-time information due to their perspective. Although satellite data capture videos from a higher perspective, street videos capture every obstacle in a city, such as motorized vehicles or construction sites, and could, therefore, offer a different measure of greenness.

*Research Objectives*

Previous studies have shown the importance of scenic routes and trees in choosing a cycle path [10,13,26–28]. The goal of this paper is twofold. First, we aim to determine if a cyclist's video footage taken just under eye level can be used to determine the greenness level—which, in this study, is measured as the percentage of street trees—of a route using software object detection, in this case, the Detectron2 library. Second, we want to know if video data might be an alternative to canopy data, especially when data are collected while riding a bicycle. Therefore, we compare eye-level video data with canopy data derived from NDVI data for the same year in Montréal—a city where a large dataset is already available. This allows us to determine the possibility of quantifying street trees with video images and use these data in other studies.

## 2. Materials and Methods

### 2.1. Study Area and Primary Data Collection

This study is based on a primary data collection using instrumented bicycles conducted on the island of Montréal in June 2019 (2 million inhabitants in 2020). This extensive mobile data collection has previously been used in recent works to analyze cycling safety, particularly dangerous overtaking [19] and conflict occurrence with motorized vehicles and

pedestrians [20]. The reader can refer to these two studies for a detailed description of this primary data collection. Briefly, 4 participants cycled on 141 predefined routes for 87 h and 1199 km. These routes were chosen to maximize the coverage of the road and cycling networks while also taking the diversity of urban micro-environments into consideration [20]. All the subjects gave their informed consent for inclusion before participating in the study. The study was conducted in accordance with the Declaration of Helsinki and the protocol was approved by the Ethics Committee of the Institut National de la Recherche Scientifique (project No. CER 19-509). Each participant was equipped with (1) a GPS watch (Garmin Forerunner 920 XT, Olathe, KA, USA) to record GPS points every second, and (2) an action camera (Garmin VIRB XE, Olathe, KA, USA) mounted on the handlebar of the bicycle to record a video of each route.

### 2.2. GIS Secondary Data on Road Network and Canopy

As described in previous studies [19,20], all GPS points were map-matched on the OpenStreetMap (OSM) [29] network data and manually validated to extract the type of road (primary, secondary, tertiary, service, residential, etc.) using the *highway* key from OSM [30] (Table 1).

**Table 1.** GPS points and road types.

| Type of Road [1] | N | % | HH:MM:SS |
|---|---|---|---|
| Primary | 14,650 | 4.70 | 04:04:10 |
| Secondary | 70,968 | 22.79 | 09:42:48 |
| Tertiary | 69,996 | 22.47 | 19:26:36 |
| Service | 2233 | 0.72 | 00:37:13 |
| Residential | 96,636 | 31.03 | 26:50:36 |
| Unclassified | 5938 | 1.91 | 01:38:58 |
| Cycleway | 40,587 | 13.03 | 11:16:27 |
| Footway | 8765 | 2.81 | 02:26:05 |
| Pedestrian | 1673 | 0.54 | 00:27:53 |
| **Total** | **311,446** | **100.0** | **86:30:46** |

[1] Based on the *highway* key from OpenStreetMap [30].

The canopy data were downloaded from the Montreal Urban Community Website [31]. Built from NDVI data and a digital height model (DHM), this open dataset contains four categories (covers): *low mineral*, *high mineral*, and *low vegetal* and *high vegetal* (canopy), where *low* is below three meters from ground level and *high* is above three meters. The difference between *low vegetal* and *high vegetal* is the NDVI value, with data lower than 0.3 being *low* and the rest being *high* (with data ranging from $-1$ to 1). With these categories, we isolate the *high vegetal* cover, creating a map of the canopy on the Island of Montréal.

### 2.3. Data Processing

The data processing was conducted entirely in Python and is illustrated in Figure 2.

First, one image per second was extracted from each video using the OpenCV library [32]. In total, 311,446 images were generated. Second, each video image was analyzed using the Detectron2 [33] library, which is implemented in PyTorch [34], in order to calculate the number of trees each frame contained. It should be noted that although this study focused on the trees, the Detectron2 algorithm can also identify flowers, grass, and other types of vegetation. The configuration used for Detectron2 was the *COCO-PanopticSegmentation* file provided by the library [35]. During this process, three other features were extracted: buildings, roads, and sky. As an example, 17.1%, 32.7%, 21.8%, and 25.1% of trees, roads/pavement, sky, and buildings, respectively, were detected in the image in Figure 3. Note that these percentages were annotated on each frame to verify whether they made sense. These percentages were also saved to a text file and their univariate statistics are reported in Table 2.

Third, the results obtained from the Detectron2 analysis were then merged with the GPS points collected on each route and saved in a geopackage file (*gpkg*). The values were associated with each point using the route filename, as well as the timestamp (DD:HH:MM:SS).

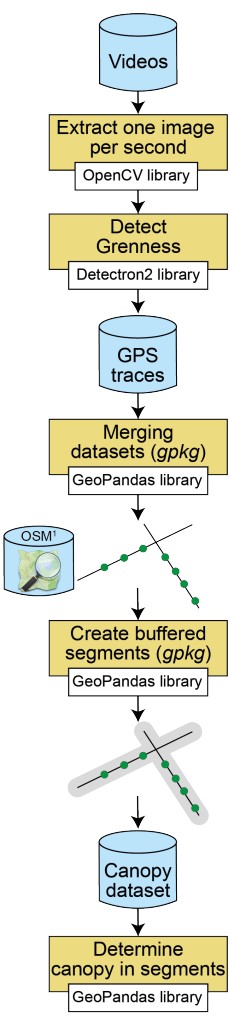

¹ OpenStreetMap (road network).

**Figure 2.** Data processing in Python.

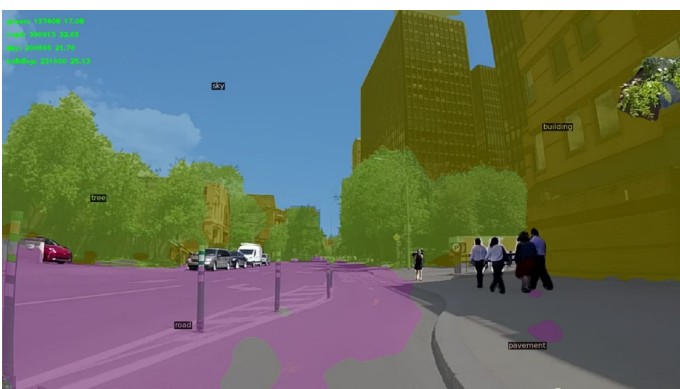

**Figure 3.** Segmented image; 17.08% of image pixels were classified as trees.

**Table 2.** Univariate statistics for the four detected categories in the images (n = 311,446).

|  | Tree | Road | Sky | Building |
|---|---|---|---|---|
| Percentiles |  |  |  |  |
| 1 | 0.0 | 0.5 | 0.0 | 0.0 |
| 5 | 0.8 | 10.6 | 1.5 | 0.0 |
| 10 | 2.3 | 18.0 | 3.2 | 0.0 |
| 25 | 7.3 | 28.6 | 8.1 | 0.0 |
| 50 | 17.5 | 39.3 | 16.3 | 2.7 |
| 75 | 29.9 | 48.0 | 26.7 | 11.2 |
| 90 | 41.8 | 55.3 | 36.9 | 22.9 |
| 95 | 49.4 | 58.9 | 42.9 | 30.1 |
| 99 | 65.8 | 65.6 | 54.2 | 42.4 |
| Mean | 20.2 | 37.8 | 18.5 | 7.5 |
| SD [1] | 15.7 | 14.4 | 13.0 | 10.3 |

[1] SD: standard deviation.

Fourth, each route was split into segments ranging from 100 to 400 m, with a step of 50 m. The different lengths were compared to see if a specific length was more efficient at predicting the canopy at the route level (i.e., sensitivity analysis). These segments were created from the GPS coordinates of each route using the GeoPandas library [36]. The segments also contained a greenness level ($g_S$), which is the weighted mean tree percent of each point ($g_i$), where $w_i$ is the distance between the point $i$ and the next point over the segment length ($l_S$):

$$g_S = \frac{\sum_{p \in S}^{i=1} g_i w_i}{\sum_{p \in S}^{i=1} w_i} \text{ with } w_i = \frac{d(i, i+1)}{l_S} \tag{1}$$

Formula (1) was used to fix the following problem: data points where the cyclist was stopped tended to accumulate because we had one point per second, but the urban features on the image hardly varied (Figure 4). By looking at the distance to the next point, we smoothed out any accumulation of points that did not provide a new value.

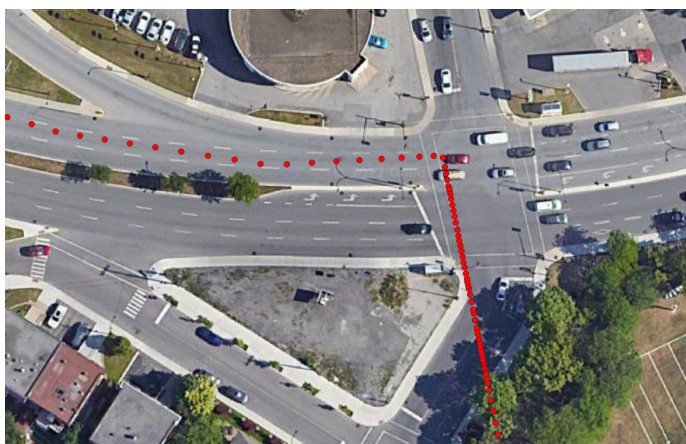

**Figure 4.** GPS points on a route with different distances between them.

We then added a buffer of 15 m on each side of each point (Figure 5a). This buffer allowed us to create polygons with each segment representing the field of view. Because we wanted to compare the greenness detected in videos with the canopy data, we limited the observable area to that seen by the camera. For example, we could not see on the other side of buildings or very far on each side. It would be time-consuming and difficult to manually enter a distance for each point based on the built environment and visual observations so a buffer of 15 m on each side was chosen. We selected this 15 m threshold because, when observed in GIS software, it reflected the average road width for the routes in Montréal. The polygons, therefore, represented the field of view on each side of the video for most of

the city. Because there were points where the buffer was too low or high, the final results might have been affected.

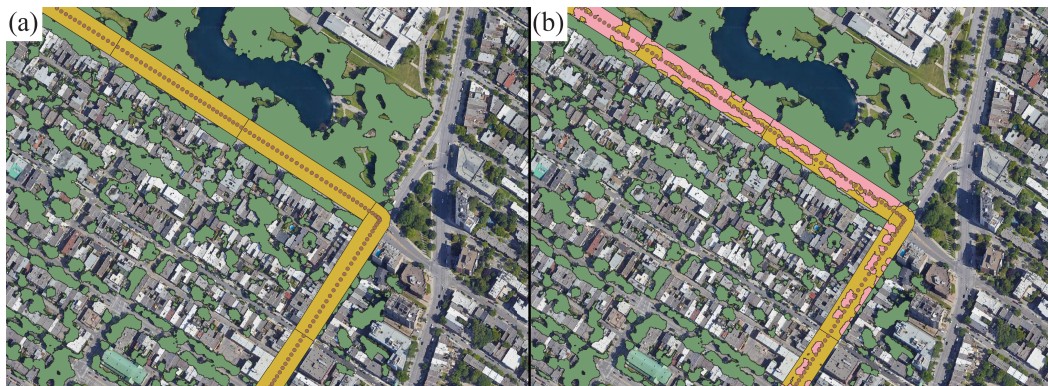

**Figure 5.** Example of road segments. (**a**) Road segment polygons. (**b**) Road segments with overlapping canopy.

Finally, it was possible to determine the parts of the canopy derived from NDVI data that intersected with the route segments and calculate a canopy area percentage ($C_S$, Equation (2)) for each segment (Figure 5b). The univariate statistics of these $C_S$ indicators are reported in Table 3.

$$C_S = \frac{\text{Area of canopy in segment}}{\text{Total area of segment}} \times 100 \quad (2)$$

**Table 3.** Univariate statistics for the percentages of the canopy within the buffered segments.

| Segment Length | 100 m | 150 m | 200 m | 250 m | 300 m | 400 m |
|---|---|---|---|---|---|---|
| n [1] | 13,978 | 9309 | 6972 | 5569 | 4625 | 3444 |
| Percentiles | | | | | | |
| 1 | 0.0 | 0.0 | 0.0 | 0.0 | 0.0 | 0.0 |
| 5 | 0.0 | 0.0 | 0.1 | 0.2 | 0.4 | 0.8 |
| 10 | 0.2 | 0.7 | 1.1 | 1.7 | 1.7 | 2.2 |
| 25 | 4.2 | 5.1 | 5.5 | 6.6 | 6.6 | 7.0 |
| 50 | 14.5 | 15.0 | 15.3 | 15.9 | 15.9 | 16.3 |
| 75 | 29.6 | 28.9 | 28.9 | 28.6 | 28.6 | 28.0 |
| 90 | 46.7 | 45.3 | 43.6 | 42.6 | 42.0 | 40.8 |
| 95 | 57.4 | 55.5 | 53.6 | 52.8 | 51.6 | 49.4 |
| 99 | 82.9 | 77.8 | 77.5 | 76.3 | 73.5 | 72.3 |
| Mean | 19.6 | 19.6 | 19.6 | 19.6 | 19.6 | 19.5 |
| SD [2] | 19.1 | 18.2 | 17.6 | 17.2 | 16.7 | 16.2 |

[1] n: number of buffered segments. [2] SD: standard deviation.

### 2.4. Data Analysis

All the statistical analyses were conducted using R (version 4.0.5) [37]. Following image segmentation (using Detectron2), two types of analyses were performed. First, a Pearson correlation matrix was built to explore the associations between the four categories (*tree*, *road*, *sky*, *building*). Second, box plots, analysis of variance (ANOVA), and the Kruskal–Wallis test by ranks were used to test whether the percentage of greenness varied with the road types identified by OpenStreetMap. This allowed us to determine if a certain road type had more greenness than others.

In line with the second objective—to verify whether video data might be an interesting alternative to canopy data for quantifying the vegetation on a given route—bivariate analyses (simple regression and correlation analyses) were performed with the greenness

indicators obtained using Detectron2 ($g_S$) and the canopy indicators ($C_S$) for the buffered segment from 100 to 400 meters.

## 3. Results

### 3.1. Tree Detection

The first results were obtained after the videos were processed to detect the components (i.e., *tree*, *road*, *sky*, *building*). The algorithm was able to detect the trees in each image quite accurately. Not all of the >300,000 images were revised by humans but each image that was observed had a correct percentage of trees identified (Figure 6). These results are in line with the literature, which shows that the Detectron2 library was accurate [24]. There were some areas, such as the second picture in the set, where small shrubs were considered trees, which might have impacted the results.

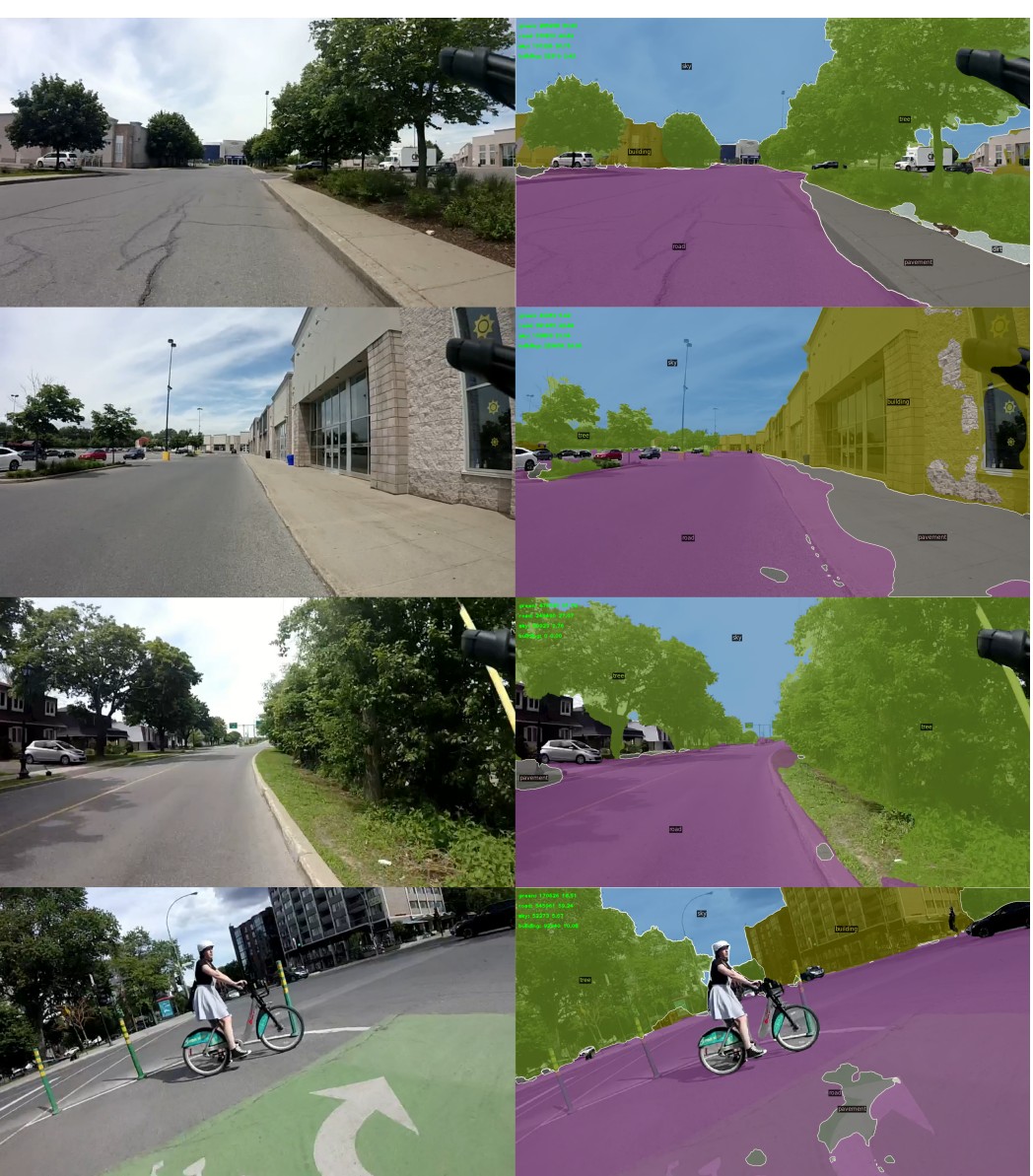

**Figure 6.** Detection of trees in different settings.

The correlation matrix between the proportions of the four categories detected in the 311,446 images is reported in Table 4. The more roads, sky, or buildings detected in the picture, the less greenness there was. The strongest negative correlation was observed between the proportions of the *tree* and *building* categories (r = −0.452, *p* < 0.001). There was also a positive correlation between the *road* and *building* categories (r = 0.113, *p* < 0.001). These results might seem obvious but they present a good argument that image detection works well. Thus, the Detectron2 algorithm can correctly determine the parts of the image that correspond to each category.

**Table 4.** Pearson correlation matrix between the proportions of the four detected categories [1].

|  | Tree | Road | Sky | Building |
|---|---|---|---|---|
| Tree |  | −0.364 | −0.416 | −0.452 |
| Road | −0.364 |  | −0.200 | 0.113 |
| Sky | −0.416 | −0.200 |  | −0.151 |
| Building | −0.452 | 0.113 | −0.151 |  |

[1] All correlation values were significant at *p* < 0.001.

Once the greenness indicators at the GPS points and buffered segments were obtained, they could be mapped with GIS software such as QGIS [38], as shown for a portion of a route in Figure 7.

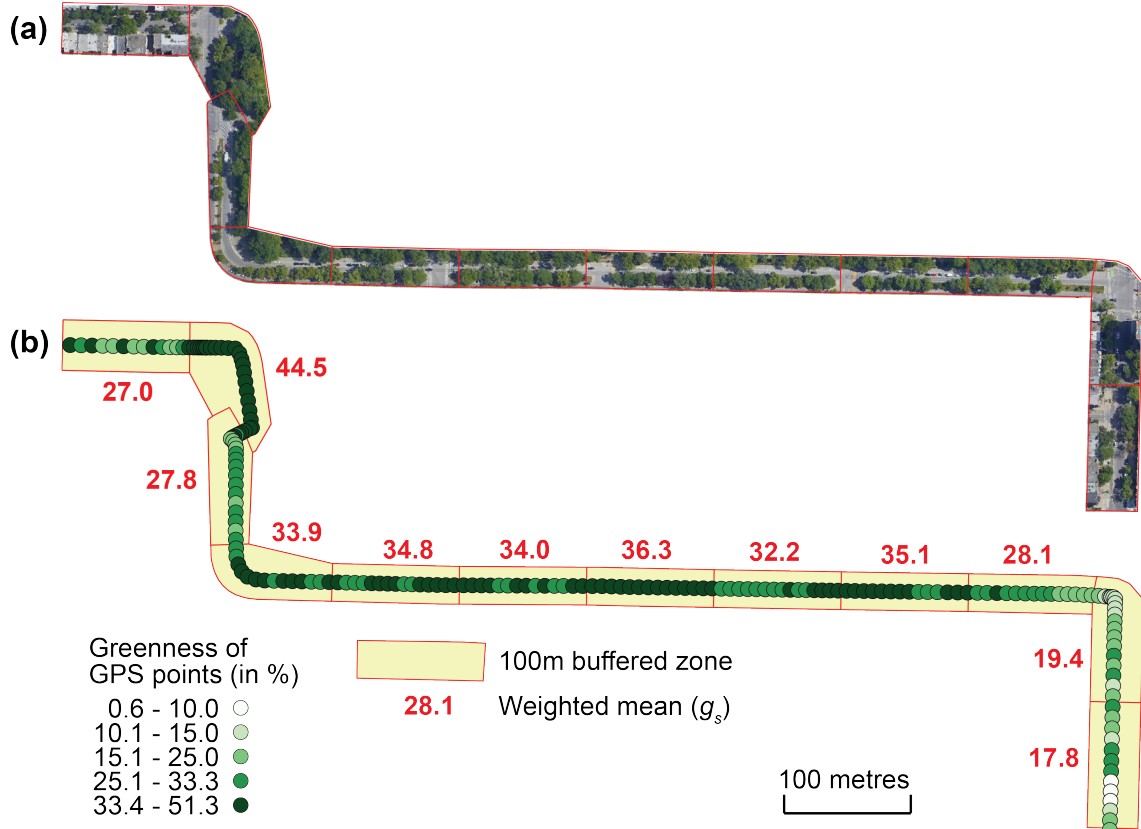

**Figure 7.** Mapping results along a route. (**a**) Google Maps imagery and 100 m buffered segment in red. (**b**) GPS points and 100 m buffered segments. The greenness of the GPS points is defined as the percentage of trees detected in the image using Detectron2.

Unsurprisingly, the percentage of greenness (trees) varied according to the road type, as illustrated by the box plots in Figure 8. The pedestrian streets had the most greenness, most of them being located in large parks (e.g., Mont-Royal Park). Primary and secondary roads had the least greenness because these roads, according to the OpenStreetMap classification, are the most important streets in the road network, most of them being larger roads and used mainly by cars. It was also a design choice by the city of Montréal, meaning that the results might differ in another city. Conversely, the presence of trees was more important on residential streets, cycleways, and pedestrian streets.

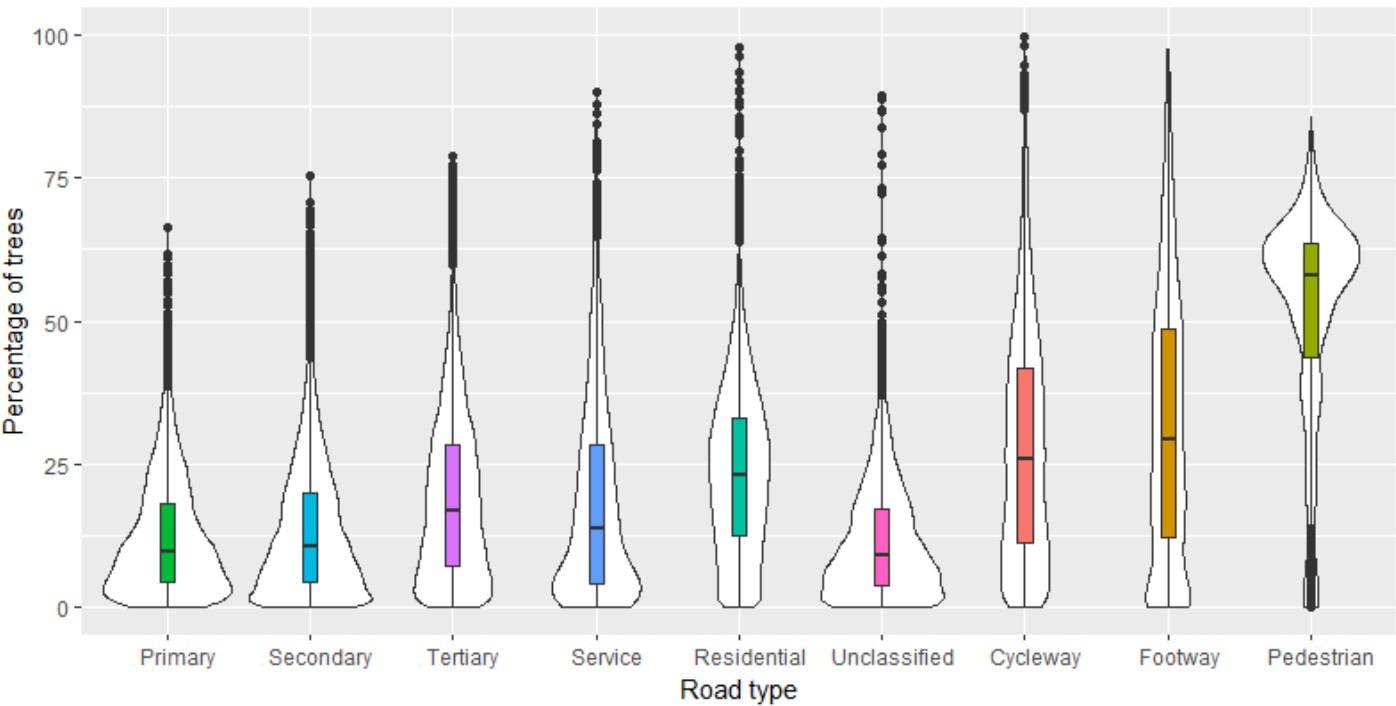

**Figure 8.** Greenness per road type (i.e., percentage of trees). ANOVA: Welch's $F(8, 311,437) = 6842$, $p < 0.001$, Eta$^2 = 0.15$. Kruskal–Wallis test: $\chi^2(8) = 39{,}112$, $p < 0.001$.

### 3.2. Correlation between Detected Greenness and Canopy Data

Simple linear regression and Pearson correlation coefficients were calculated to assess the relationship between the percentage of trees detected in the images and the percentage of canopy, with segments of 100, 150, 200, 250, 300, and 400 m (Figure 9). All the correlation coefficient values were significant ($p < 0.001$) and varied slightly between 0.77 and 0.82. The sensitivity analysis showed that the segment length did not have a significant effect on the correlation between the video greenness (tree) and canopy coverage.

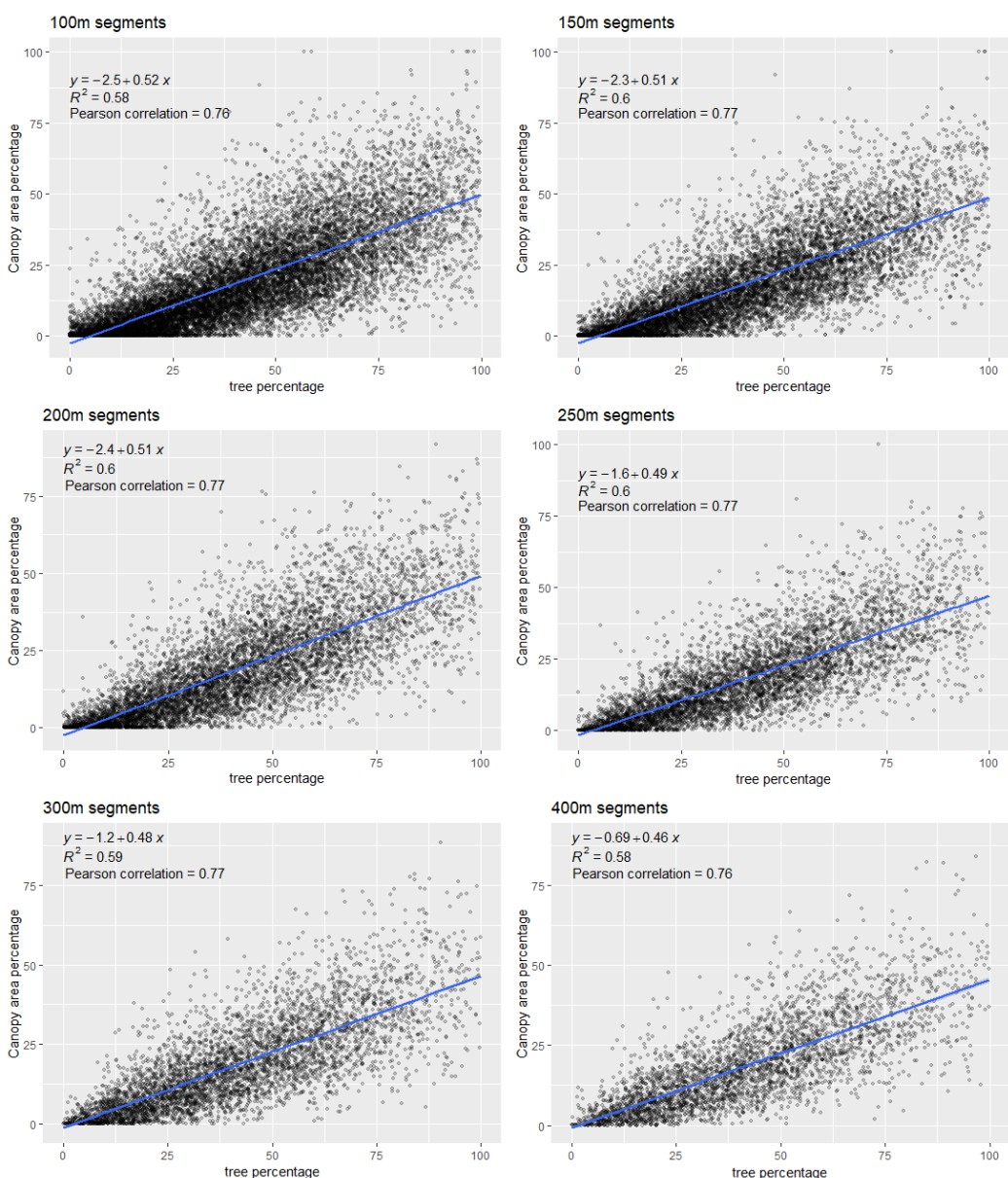

**Figure 9.** Greenness and canopy correlation.

## 4. Discussion

### 4.1. Using Detectron2 to Detect Greenness

Using Detectron2 to detect trees works well according to the pictures created during the generation of the greenness index. This indicates that it is possible to use videos collected while riding a bicycle in cities to determine the number of street trees on a route. This approach is particularly relevant for cities and towns that do not possess canopy data—data that are often used in scientific research to determine greenness. This approach could also be used to detect other urban features (e.g., roads, buildings, sidewalks), urban objects (traffic lights, fire hydrants, street signs, stop signs, parking meters, benches), and street users (motorized vehicles, pedestrians, cyclists). This is particularly relevant for cities where Open GIS Data, Google Street View images, or satellite imagery are not available. Moreover, because the videos are taken roughly at eye level, they better represent the scenery observed while riding a bicycle in the city. It is easier to see the number of cars on the street, which might hide some of the vegetation, rendering the route less green and therefore less scenic (Figure 10).

The greenness index created using the videos represents a more natural way of determining greenness because the videos are taken at eye level. The videos were taken on a certain day and time, meaning it is unlikely that the same route would return the same value on another day. In other words, future works could explore how the amount of vegetation visible to a cyclist could vary according to the time and day (e.g., rush hour versus the rest of the day, weekdays versus weekends) or the season for the same route. This is an advantage because eye-level greenness is more representative of the scenery than satellite images.

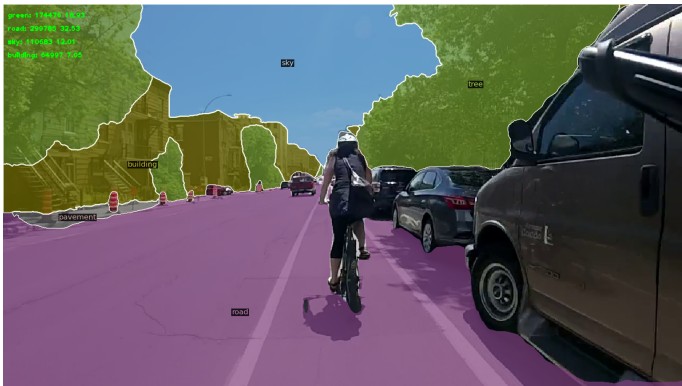

**Figure 10.** Motorized vehicle blocking trees.

### 4.2. Comparing Canopy Data to Video Greenness

The strong correlation values (>0.75) obtained between the canopy data indicators and video greenness demonstrate that the proposed approach to evaluating trees using software object detection in videos is relevant. This finding should be validated in other cities, particularly European cities and cities in the Global South, where the amount and species of street trees could be very different.

However, two principal elements could explain the discrepancy between the two datasets. The first is related to the chosen buffer width (15 m) representing the field of view. Ideally, we would have the exact width of each street. Unfortunately, open GIS data on street widths are not available for the study area, which is also true for many cities around the world. The second is that the two indicators measured two different features: (1) the canopy as seen from the sky, and (2) trees as seen from the street, with different obstacles obstructing the view (e.g., motorized vehicles). Although there was a strong correlation between the two, both could be used as distinctive variables in studies observing the different impacts of greenness.

### 5. Conclusions

To conclude, this paper had two goals: (1) to determine whether videos recorded by a camera fixed on a bicycle's handlebar can be used to determine greenness, and (2) to determine if videos can replace canopy data. For the first goal, we found that by using Detectron2, we could accurately detect trees in images taken at eye level. The percentage of trees (and other categories such as flowers and grass [35]) offers a more realistic view since it records the same view as the cyclist including road obstacles. This may help in calculating a more representative greenness level in different cities, especially when including all types of greenness in the detection algorithm. For the second goal, strong correlations were found between the two types of vegetation indicators. This means that canopy data could be replaced by video-detected greenness. This finding can, therefore, be useful for future urban mobility studies that already use cameras and take into consideration a user's perspective of vegetation. It could also be applied to other aspects of the built environment such as buildings and roads.

**Author Contributions:** Conceptualization, Albert Bourassa, Philippe Apparicio, Jérémy Gelb, and Geneviève Boisjoly; methodology, Albert Bourassa, Philippe Apparicio, and Jérémy Gelb; software, Albert Bourassa; data validation, Albert Bourassa, and Philippe Apparicio; statistical analyses, Albert Bourassa, and Philippe Apparicio; writing—original draft preparation, Albert Bourassa and Philippe Apparicio; writing—review and editing, Albert Bourassa, Philippe Apparicio, Jérémy Gelb, and Geneviève Boisjoly; supervision, project administration, funding acquisition, Philippe Apparicio. All authors have read and agreed to the published version of the manuscript.

**Funding:** This study was financially supported by the Social Sciences and Humanities Research Council of Canada (SSHRC) through an Insight Grant (435-2019-0796).

**Institutional Review Board Statement:** This study has been approved by the Research Ethics Board of the Institut National de la Recherche Scientifique (project No. CER 19-509, date of approval: 28 May 2019).

**Informed Consent Statement:** Informed consent was obtained from all the subjects involved in the study.

**Data Availability Statement:** The Python code to evaluate the canopy detection with Detectron2 is available at: https://gitlab.com/albert.bourassa/detectron2_canopy (accessed on 24 August 2022).

**Acknowledgments:** Special thanks to the four cyclists who were involved in the data collection.

**Conflicts of Interest:** The authors declare no conflict of interest.

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
