# Peer review of "Canopy Assessment of Cycling Routes: Comparison of Videos from a Bicycle-Mounted Camera and GPS and Satellite Imagery"

_ijgi, doi:10.3390/ijgi12010006_

Round 1

Reviewer 1 Report

This study determines if video data can access tree canopy on bicycle routes. The study also compares the ability to assess bicycle route tree canopy using GPS with a camera-mounted bicycle versus using satellite imagery (normalized difference vegetation index NDVI). The GPS with a camera-mounted bicycle could be more affordable and accurate because the satellite’s high-resolution imagery is costly to obtain and cloud cover sometimes blocks the view.

Title: The title needs changing because “greenness” is not the measure assessed by the authors.  Greenness includes lawns, flowers, and bushes and the authors are comparing tree canopy using two different methods.  The word “scenic” also does not belong because scenic can include views of rivers, valleys, historic buildings, cows, horses, or farm fields.  Cycling routes can remain because those are the locations for the trees. The word assessment can remain but the reader will not know the meaning of Detectron2 Algorithm.  That is the tool used to calculate the number of trees in each frame and its use is not being studied.  Consider this title, “Canopy Assessments on Cycling Routes: Accuracy and cost comparisons of GPS with a camera mounted bicycle versus satellite imagery.”

Abstract:  The abstract needs to be rewritten to provide clarity for the reader.  The measure is not beautiful scenery and the assessment is not greenness.  There is one assessment, i.e., tree canopy, one location, cycle routes, and two tools for comparison: 1) GPS with a camera mounted bike and 2) satellite imagery.   Mention of “obstacles” is confusing because the route is a bicycle route and obstacles are in the route? Where are these obstacles?  The abstract should include the text on lines 27-30. The abstract should also include the text on lines 37 to 40.  The authors decided to not compare Google Street View and the Google View Index with the GPS and camera mounted bicycle because Google Street View data includes streets in winter and not all cities have Google Street View. The authors did decide to compare GPS with the camera mounted bicycle versus the satellite imagery because, though costly, cities do have access to the satellite imagery.  (The reader will be extremely pleased to know that the authors are exploring a more affordable, full leaf, and accurate tool to assess the tree canopy on bicycle routes. The authors need to convey the care for others in this study. The study is extremely worthwhile!)  On lines 215 to 216, the authors present an additional goal. This should be in the Abstract also but changed to: “1) To determine if video data can be used to determine tree canopy; and 2) to determine if video data is superior in cost and accuracy to canopy data from satellite imagery.   

Introduction: The measure is of trees so the references should include papers that assessed preferences about trees on bicycle routes. The below are some articles to consider citing:

 “Bicycle Level of Service Model for the Cycloruta, Bogota, Columbia””

“Pedestrian and cyclists preferences for tree locations by sidewalks and cycle tracks and associated benefits: Worldwide implications from a study in Boston, MA”.

“A Theoretical Perspective on How Bicycle Commuters Might Experience Aesthetic Features of Urban Space”

     The authors can also include the value of trees for addressing heat island:

“Scale-dependent interactions between tree canopy cover and impervious surfaces reduce daytime urban heat during summer”0)

These articles were found by typing “tree” and “bicycle” into Google Scholar.

   The authors need to convince the readers of the contribution of their study and the value of trees, especially on bicycle routes. That is possible through the literature in the Introduction. The authors need to watch parallels, “Indeed, tree canopy is an element of nature which contributes to positive emotions and therefore to nicer scenery.”  Emotions and nicer scenery are not parallel elements though nicer scenery does improve mood (and directed attention fatigue). Here is another citation. “Perceived green at speed: a simulated driving experiment raises new questions for attention restoration theory and stress reduction theory.” The trees are viewed at the speed of a bicyclist. 

Throughout the remainder of the Introduction, the authors need to keep repeating only what is studied.  The authors do need to explain eye level when the camera will be on the handlebars of the bike. This is not eye level.

Results:   On line 167, the authors would not write “Unsuprisingly.”  The authors can just present the findings.  Also, on line 167, the authors write “pedestrian street” but this study is about bicycle routes.  If the authors also want to include pedestrian routes, this should be in the title, Abstract, and Results. 

On line 189, mention is made of the number of cars on the street and hiding some vegetation.  How can cars hide a tree canopy?  The authors should also not write, “In other words” like 195.

Line 211 is confusing because 2) trees as seen from the street with different obstacles obstructing the view. The reader still hasn’t been told about obstacles. 

Conclusion: The conclusion text about other greenery captured, including flowers and grass.  The authors return to the measure of “greenness?” (line 220) and the measure in this study is of trees. As written in the text about the Abstract, the text in lines 215 and 216 should be in the Abstract. "1) To determine if video data can be used to determine greenness and 2) to determine if video can replace canopy data. Again, clarify the writing to help the reader and also to give the world a better measure. With climate change, trees are greatly needed and all measures that show how to assess trees will help.          

Author Response

Dear Editors,
Thank you for giving us the opportunity to review the manuscript. We also thank the reviewers for their comments and suggestions on the first draft of the manuscript. The following text contains their comments and the responses in red that we have given to the article. Please note that in this new version the changes have been highlighted in yellow.

---------------------------

1. Title: The title needs changing because “greenness” is not the measure assessed by the authors. Greenness includes lawns, flowers, and bushes and the authors are comparing tree canopy using two different methods. The word “scenic” also does not belong because scenic can include views of rivers, valleys, historic buildings, cows, horses, or farm fields. Cycling routes can remain because those are the locations for the trees. The word assessment can remain but the reader will not know the meaning of Detectron2 Algorithm. That is the tool used to calculate the number of trees in each frame and its use is not being studied.  Consider this title, “Canopy Assessments on Cycling Routes: Accuracy and cost comparisons of GPS with a camera mounted bicycle versus satellite imagery.”

Response: Thanks for your title suggestion. We do not think it would be useful to include the notions of cost and accuracy in the title. The article doesn’t really do a cost analysis of both detection methods. We compare the two methods by using correlation analysis, but we do not determine which method is the most accurate. Consequently, the title has been modified as follows: “Canopy Assessment on Cycling Routes: Comparison of a Camera Mounted Bicycle with GPS Versus Satellite Imagery”.

2. Abstract: The abstract needs to be rewritten to provide clarity for the reader.  The measure is not beautiful scenery and the assessment is not greenness.  There is one assessment, i.e., tree canopy, one location, cycle routes, and two tools for comparison: 1) GPS with a camera mounted bike and 2) satellite imagery.   Mention of “obstacles” is confusing because the route is a bicycle route and obstacles are in the route? Where are these obstacles?  The abstract should include the text on lines 27-30. The abstract should also include the text on lines 37 to 40.  The authors decided to not compare Google Street View and the Google View Index with the GPS and camera mounted bicycle because Google Street View data includes streets in winter and not all cities have Google Street View. The authors did decide to compare GPS with the camera mounted bicycle versus the satellite imagery because, though costly, cities do have access to the satellite imagery.  (The reader will be extremely pleased to know that the authors are exploring a more affordable, full leaf, and accurate tool to assess the tree canopy on bicycle routes. The authors need to convey the care for others in this study. The study is extremely worthwhile!)  On lines 215 to 216, the authors present an additional goal. This should be in the Abstract also but changed to: “1) To determine if video data can be used to determine tree canopy; and 2) to determine if video data is superior in cost and accuracy to canopy data from satellite imagery.

Response: To address these comments, these modifications have been done: 1) ’urban scenery’ has been replaced by ‘urban greenness’, 2) we remove Google Street View.  While we do mention the cost of satellite data, we do not include a cost comparison because it would be difficult to do depending on the equipment and labours costs to collect data in different cities. The objectives mentioned on lines 215-216 are the same as the ones already in the abstract but formulated a bit differently. In other words, we evaluate the correlation between the two methods, but we do not compare which method is superior in terms of detection accuracy of canopy.

3. Introduction: The measure is of trees so the references should include papers that assessed preferences about trees on bicycle routes. The below are some articles to consider citing:

 “Bicycle Level of Service Model for the Cycloruta, Bogota, Columbia”

“Pedestrian and cyclists preferences for tree locations by sidewalks and cycle tracks and associated benefits: Worldwide implications from a study in Boston, MA”.

“A Theoretical Perspective on How Bicycle Commuters Might Experience Aesthetic Features of Urban Space”

Response: The first article suggested is not adequate from our perspective. We don’t think that it would be a great addition. The other two have been added to the introduction to make the literature review better. Thank you for those suggestions.

4. The authors can also include the value of trees for addressing heat island: “Scale-dependent interactions between tree canopy cover and impervious surfaces reduce daytime urban heat during summer”

Response: In our opinion, this fact is already well known in transportation and planning circles, and it is outside of the scope of our study.

5. These articles were found by typing “tree” and “bicycle” into Google Scholar. The authors need to convince the readers of the contribution of their study and the value of trees, especially on bicycle routes. That is possible through the literature in the Introduction. The authors need to watch parallels, “Indeed, tree canopy is an element of nature which contributes to positive emotions and therefore to nicer scenery.” Emotions and nicer scenery are not parallel elements though nicer scenery does improve mood (and directed attention fatigue). Here is another citation. “Perceived green at speed: a simulated driving experiment raises new questions for attention restoration theory and stress reduction theory.” The trees are viewed at the speed of a bicyclist.

Response: The sentence quoted has been modified and a suggested article was added as an argument to show that trees contribute to positive emotions, removing the parallel and creating two separate arguments. The study on perceived green at speed is out of the scope of our paper. This interesting study has been conducted in lab environment simulating cars driving on the freeway in Hong Kong. Consequently, their results could not be applied to cyclists.  

6. Throughout the remainder of the Introduction, the authors need to keep repeating only what is studied. The authors do need to explain eye level when the camera will be on the handlebars of the bike. This is not eye level.

Response: A phrase was added to be more precise about the camera location (lines 52-53).

7. Results: On line 167, the authors would not write “Unsuprisingly.”  The authors can just present the findings.  Also, on line 167, the authors write “pedestrian street” but this study is about bicycle routes.  If the authors also want to include pedestrian routes, this should be in the title, Abstract, and Results

Response: The term has been removed.

8. On line 189, mention is made of the number of cars on the street and hiding some vegetation. How can cars hide a tree canopy? 

Response: This sentence is linked to Figure 11, where we see a car that is hiding some parts of the trees on the right side of the image.

9. The authors should also not write, “In other words” like 195.

Response: The term on line 195 was removed.

10. Line 211 is confusing because 2) trees as seen from the street with different obstacles obstructing the view. The reader still hasn’t been told about obstacles.

Response: A phrase was added in the intro to mention what obstacles we are referring to (lines 54-55). The sentence has been modified as follows: “2) trees as seen from the street, with different obstacles obstructing the view (e.g. motorized vehicles)”.

11. Conclusion: The conclusion text about other greenery captured, including flowers and grass. The authors return to the measure of “greenness?” (line 220) and the measure in this study is of trees. As written in the text about the Abstract, the text in lines 215 and 216 should be in the Abstract. "1) To determine if video data can be used to determine greenness and 2) to determine if video can replace canopy data. Again, clarify the writing to help the reader and also to give the world a better measure. With climate change, trees are greatly needed and all measures that show how to assess trees will help.

Response: The abstract was modified to talk more about trees. In the section titled Research Objectives, we mention that the greenness level in our study is “measured as the percentage of street trees”. We feel that the lines 215-216 are already in the introduction, although formulated a tad differently.

Reviewer 2 Report

An interesting and well-written paper, the authors assessed the greenness level of cycling routes by applying the software objective detections on video and comparing the detected greenness data to the canopy data. I only suggest one minor revision to the manuscript for the authors to consider. It would be better if the authors could discuss more how their research contributes to the current urban studies.

Author Response

Dear Editors,
Thank you for giving us the opportunity to review the manuscript. We also thank the reviewers for their comments and suggestions on the first draft of the manuscript. The following text contains their comments and the responses in red that we have given to the article. Please note that in this new version the changes have been highlighted in yellow.

---------------------------

Response: Thank you for your encouragement. As mentioned in the first paragraph of the discussion section, “This approach is particularly relevant for cities and towns that do not possess canopy data, data which is often used in scientific research to determine greenness”. We also added one sentence: “Moreover, this approach could be used to detect other urban features (e.g. road, building, sidewalk), urban objects (traffic light, fire hydrant street sign, stop sign, parking meter, bench) and street users (motorized vehicles, pedestrians and cyclists). This is particularly relevant for cities where Open GIS Data, street view images or satellite imagery are not available.” (lines 193-197).

Reviewer 3 Report

This paper attempts to compare the eye-level greenness data obtained by semantic segmentation of actual street view with the tree canopy data set obtained by NDVI, to discuss whether street green visual rate data can replace tree canopy data.

A key issue to this paper is its originality. There is a paper recently published by International Journal of Geo-Information (shown as below). I didn't see this reference in you reference list. It also compared the eye-level  greenness obtained by Street View segmentation with NDVI.

Gao, F.; Li, S.; Tan, Z.; Zhang, X.; Lai, Z.; Tan, Z. How Is Urban Greenness Spatially Associated with Dockless Bike Sharing Usage on Weekdays, Weekends, and Holidays? ISPRS Int. J. Geo-Inf. 2021, 10, 238. https://doi.org/10.3390/ijgi10040238

Secondly, I have questions about the research design of this paper:Is the correlation between green vision and canopy data related to vertical structure of local vegetation community?

In a place where the plant biomass is concentrated in the tree canopy, the eye-level greenness can be substituted for the tree canopy data. But if the plant biomass in a place is evenly distributed among the tree, shrub, herb, and ground layers, greenness is not representative of the canopy. The street view data used by the research institute came from Montreal, Canada, which belongs to the temperate continental humid climate. The urban plant species are mainly trees, the vertical level of vegetation is less, and the plant biomass is more concentrated in the tree canopy. In subtropical regions, however, the vegetation biomass of low-growing plants is greater. In this paper, semantic segmentation classified all plant types as trees, which may have a great impact on the research if the vertical structure of vegetation community is relatively complex in the area. The researchers also mention in line 153 that shrubs can be misclassified as trees. In my opinion, if the tree canopy cannot be correctly classified, the reliability and universality of the method in this paper is not good enough. In my opinion, researchers should try to improve the semantic segmentation of this paper to improve the classification accuracy of plants, or conduct a comparative study of cities in different climate zones to prove the universality of the study.

Third, this paper has little research content. It has spent a long time to describe data preprocessing and few contents of results, discussions and conclusions.

Fourthly, most of the drawings in the article are diagrams of the processing process, which can be made into a group of drawings to reduce the number of drawings.

Author Response

Dear Editors,
Thank you for giving us the opportunity to review the manuscript. We also thank the reviewers for their comments and suggestions on the first draft of the manuscript. The following text contains their comments and the responses in red that we have given to the article. Please note that in this new version the changes have been highlighted in yellow.

---------------------------

1. This paper attempts to compare the eye-level greenness data obtained by semantic segmentation of actual street view with the tree canopy data set obtained by NDVI, to discuss whether street green visual rate data can replace tree canopy data. A key issue to this paper is its originality. There is a paper recently published by International Journal of Geo-Information (shown as below). I didn't see this reference in you reference list. It also compared the eye-level greenness obtained by Street View segmentation with NDVI.
Gao, F.; Li, S.; Tan, Z.; Zhang, X.; Lai, Z.; Tan, Z. How Is Urban Greenness Spatially Associated with Dockless Bike Sharing Usage on Weekdays, Weekends, and Holidays? ISPRS Int. J. Geo-Inf. 2021, 10, 238. https://doi.org/10.3390/ijgi10040238

Response: Thanks for this reference. The study mentioned was added to the argument saying that GSV is used to measured greenness. The text has been modified as follows: “An alternative is to analyse street view images – obtained from Google Street View (GSV) or BMap – which provide a visualization of vegetation as it might be seen by an individual on the street [13–16]” (lines 32-34).

2. Secondly, I have questions about the research design of this paper: Is the correlation between green vision and canopy data related to vertical structure of local vegetation community? In a place where the plant biomass is concentrated in the tree canopy, the eye-level greenness can be substituted for the tree canopy data. But if the plant biomass in a place is evenly distributed among the tree, shrub, herb, and ground layers, greenness is not representative of the canopy. The street view data used by the research institute came from Montreal, Canada, which belongs to the temperate continental humid climate. The urban plant species are mainly trees, the vertical level of vegetation is less, and the plant biomass is more concentrated in the tree canopy. In subtropical regions, however, the vegetation biomass of low-growing plants is greater. In this paper, semantic segmentation classified all plant types as trees, which may have a great impact on the research if the vertical structure of vegetation community is relatively complex in the area. The researchers also mention in line 153 that shrubs can be misclassified as trees. In my opinion, if the tree canopy cannot be correctly classified, the reliability and universality of the method in this paper is not good enough. In my opinion, researchers should try to improve the semantic segmentation of this paper to improve the classification accuracy of plants, or conduct a comparative study of cities in different climate zones to prove the universality of the study.

Response: We restructured some of the introduction to better reflect that in this study, greenness means tree canopy. We also mention that other types of vegetation can be detected by the Detectron2 algorithm, meaning that low growing biomass can be detected. Because shrubs can be seen as small trees, and because we ultimately want to detect greenness as seen from a cyclist’s perspective, we do not think that this is a problem. Concerning the universality of the approach, we have mentioned: “This finding should be validated in other cities, particularly European and South cities where the amount and species of street trees could be very different”.

3. Third, this paper has little research content. It has spent a long time to describe data preprocessing and few contents of results, discussions and conclusions.

 Response: We do not agree with this comment. First, the main contribution of the study is methodological which explains why the Materials and Methods Section is long. Also, this section is long (5 pages) because it is illustrated by several Figures. Finally, to allow the replicability of our methodological approach, it must be described adequately. Second, we do not think the results, discussion and conclusion section is too short (6 pages).

4. Fourthly, most of the drawings in the article are diagrams of the processing process, which can be made into a group of drawings to reduce the number of drawings.

Response: We have merged the figure 5 and 6 of the first version submitted.

Round 2

Reviewer 1 Report

Title: Excellent!  Note that in the title the first tool cited is the 1) Camera Mounted Bicycle with GPS and the second is the 2) Satellite Imagery.  To help the reader, any text about these two tools should remain in the same order.

Abstract: With climate change, the goal is not just to “assess the quality” but to anticipate that more bicyclists would use a route with a tree canopy.  “Tools that detect trees provide data to support maintaining or designing cycling routes to increase ridership.” The abstract starts with the Satellite Imagery and compares what exists (Satellite) to the new tool (Camera mounted bicycle) but the order in the title should remain consistent throughout the article (video camera first). These should be switched to help the reader.  Some sentences would then need to be slightly altered to help with the flow.  The third sentence ends with the “drawbacks of satellite data” but then the reader is left hanging. They will want to be convinced that the new tool is better.  The next sentence does not need the inclusion of “At the same time…”  This next sentence also is the only sentence in the Abstract to use the word “we.”  Therefore, in making the switch suggested above (camera first), the first part would be, “The rise of video recordings in data collection provides access to a new point of view of the city, data at eye level. This method may be superior to the normalized difference vegetation index (NDVI) from satellite imagery because the satellite images are costly to obtain and cloud cover sometimes obscures the view.”

Introduction:  Readers sometimes scan and it is helpful to remind them occasionally that the Detectron 2 is the software that detects the trees in the video images.  The reader will remember that this study is comparing the satellite images with video images from a camera mounted on the bicycle’s handlebars but then might not remember if the Detectron 2 is the same as the NDVI. Lines 28-55 explain this.

In lines 101-104, it would be helpful to the reader to remind the reader that Detectron 2 analyzes the video data.  “Second, each “video” image is then analyzed…”

Discussion. Line 199. It would be helpful to write “Open GIS Data, “Google Street View images …”  This was explained on lines 32-34. This will remind the reader that street view images are the Google Street View images (and not the camera mounted images…which happen to be street view).

Conclusion. Line 229  If the reader only reads the Conclusion, they may not know that Detectron2 detects the images in the bicycle mounted videos. “Using the software Detectron2 that detects the images in the bicycle mounted videos…”Line 229 should also switch two words. 
“…Detectron2, we can accurately detect trees…”

The added citations and the altered sentences made by the authors greatly improve the article.  Lines 26-27. The two benefits (positive emotions and nicer scenery) are now parallel. Also, the article now explains the obstructions.

Well done.  Only minor suggestions now to help the reader.   

Author Response

Dear Editors,

Thank you for giving us the opportunity to review the manuscript. We also thank the reviewers for their comments and suggestions on the first draft of the manuscript. The following text contains their comments and the responses in red that we have given to the article. Note the manuscript was submitted to MDPI for English editing and has been edited.

------------------------------------------------------

1. Title: Excellent! Note that in the title the first tool cited is the 1) Camera Mounted Bicycle with GPS and the second is the 2) Satellite Imagery.  To help the reader, any text about these two tools should remain in the same order.
Abstract: With climate change, the goal is not just to “assess the quality” but to anticipate that more bicyclists would use a route with a tree canopy.  “Tools that detect trees provide data to support maintaining or designing cycling routes to increase ridership.” The abstract starts with the Satellite Imagery and compares what exists (Satellite) to the new tool (Camera mounted bicycle) but the order in the title should remain consistent throughout the article (video camera first). These should be switched to help the reader.  Some sentences would then need to be slightly altered to help with the flow.  The third sentence ends with the “drawbacks of satellite data” but then the reader is left hanging. They will want to be convinced that the new tool is better.  The next sentence does not need the inclusion of “At the same time…”  This next sentence also is the only sentence in the Abstract to use the word “we.”  Therefore, in making the switch suggested above (camera first), the first part would be, “The rise of video recordings in data collection provides access to a new point of view of the city, data at eye level. This method may be superior to the normalized difference vegetation index (NDVI) from satellite imagery because the satellite images are costly to obtain and cloud cover sometimes obscures the view.”

Response: Thank you for your comments. We applied the same order in the abstract as in the title. We also added a phrase, as suggested, to drive further the point as to why tree detection is important. The new abstract reads as such: “The rise of video recordings in data collection provides access to a new point of view on the city, with data recorded at eye level. This method may be superior to the commonly used normalized difference vegetation index (NDVI) from satellite imagery because satellite images are costly to obtain and cloud cover sometimes obscures the view.”

2. Introduction: Readers sometimes scan and it is helpful to remind them occasionally that the Detectron 2 is the software that detects the trees in the video images.  The reader will remember that this study is comparing the satellite images with video images from a camera mounted on the bicycle’s handlebars but then might not remember if the Detectron 2 is the same as the NDVI. Lines 28-55 explain this.

Response: Detectron2 is mentioned on line 48 and 49. Lines 29-30 already mention NDVI is obtained from satellite images. We also think readers of the International Journal of Geo-Information (a journal of the International Society for Photogrammetry and Remote Sensing) are familiar with the NDVI method.

3. In lines 101-104, it would be helpful to the reader to remind the reader that Detectron 2 analyzes the video data. “Second, each “video” image is then analyzed…”

Response: the text has been modified as follows: “Second, each video image was analyzed using the Detectron2…”.

4. Line 199. It would be helpful to write “Open GIS Data, “Google Street View images …” This was explained on lines 32-34. This will remind the reader that street view images are the Google Street View images (and not the camera mounted images…which happen to be street view).

Response: We added “Google” to make the difference clearer.

5. Line 229 If the reader only reads the Conclusion, they may not know that Detectron2 detects the images in the bicycle mounted videos. “Using the software Detectron2 that detects the images in the bicycle mounted videos

Response: We have instead modified the previous sentence as follows: “1) to determine whether videos recorded by a camera fixed on a bicycle's handlebar can be used to determine greenness”.

6. …”Line 229 should also switch two words. “…Detectron2, we can accurately detect trees…”

Response: Thank you for noting it. We have switched these two words.

7. The added citations and the altered sentences made by the authors greatly improve the article. Lines 26-27. The two benefits (positive emotions and nicer scenery) are now parallel. Also, the article now explains the obstructions.

Response: Thanks for your encouragement and your rigorous reading of our manuscript!

Reviewer 3 Report

The authors have improved the manuscipt. 

Author Response

The authors have improved the manuscipt. Extensive editing of English language and style required.

Response: Thank you for your encouragement. As mentioned in the first paragraph of the discussion section, “This approach is particularly relevant for cities and towns that do not possess canopy data, data which is often used in scientific research to determine greenness”. We also added one sentence: “This approach could also be used to detect other urban features (e.g., road, building, sidewalk), urban objects (traffic light, fire hydrants, street sign, stop sign, parking meter, bench), and street users (motorized vehicle, pedestrian and cyclist). This is particularly relevant for cities where Open GIS Data, Google Street View images or satellite imagery are not available.” (lines 193-197).

Note the manuscript was submitted to MDPI for English editing and has been edited.